# Effect of the COVID-19 pandemic on drug-resistant tuberculosis treatment outcomes at a national referral hospital in Sierra Leone, 2017 to 2022: A retrospective cohort study

Josephine Amie Koroma[1], Mariama Mahmoud[1], Bailah Molleh[2], Stephen Sevalie[2,3,4], Adrienne K. Chan[5], Sharmistha Mishra[5], Sulaiman Lakoh[1,2,3]*, Joseph Sam Kanu[3,6]

1 Ministry of Health, Government of Sierra Leone, Freetown, Sierra Leone, 2 Research and Scientific Division, Sustainable Health Systems, Freetown, Sierra Leone, 3 College of Medicine and Allied Health Sciences, University of Sierra Leone, Freetown, Sierra Leon, 4 34 Military Hospital, Republic of Sierra Leone Armed Forces, Freetown, Sierra Leone, 5 Division of Infectious Diseases, Department of Medicine, University of Toronto, Toronto, Ontario, Canada, 6 National Public Health Agency, Government of Sierra Leone, Freetown, Sierra Leone

* lakoh2009@gmail.com

## Abstract

Sierra Leone is one of the 30 high TB burden countries in the world, with an incidence rate in 2023 of 273 per 100,000 population. Despite progress in case notification and treatment coverage, around 5,000 cases of TB in Sierra Leone are missing each year. The COVID-19 pandemic has further compounded these challenges. We highlight its effect on drug-resistant TB treatment outcomes. We conducted a retrospective cohort study of drug-resistant TB cases using national data from January 2017 to December 2022. Data was analyzed using STATA. Descriptive analysis summarized demographic, clinical characteristics and treatment outcomes. Logistic regression examined the association between time-period and outcomes, adjusting for age, gender, nutritional status, HIV status and regimen. Of 701 patients, 383 (54.6%) were registered pre-COVID-19, 228 (32.5%) during, and 92 (13.1%) post-COVID-19. Pre-treatment TB cases reduced from 359 (92.5%) in the pre-COVID-19 period to 80 (30.9%) in the COVID-19 period. New treatment cases increased from 29 (7.5%) to 159 (61.4%) during COVID-19. Treatment completion decreased from 74.7% pre-COVID-19 to 63.3% during and 68.5% post-COVID-19). Malnourished patients had higher odds of success (aOR: 1.482, 95% CI: 1.007-2.183), while those on short regimen had lower odds (aOR: 0.51, 95% CI: 0.321-0.810). We observed a decline in drug-resistant TB treatment success rate during COVID-19, which was primarily influenced by concurrent shifts in treatment protocols and underlying secular trends. The pandemic itself did not emerge as an independent determinant of poor treatment outcomes, highlighting the resilience of the TB care system. Nonetheless, the pandemic had significant indirect consequences, including worsening rates of malnutrition and HIV co-infection. These trends point to deeper systemic vulnerabilities, such as weak social protection mechanisms, increased food insecurity,

**Data availability statement:** The datasets used and/or analyzed for this study are available from: https://doi.org/10.6084/m9.figshare.30986815.

**Funding:** This research was funded by the Canadian Institutes for Health Research (CIHR) through an Operating Grant (WI1-179883) Addressing the Health Impacts of COVID-19. SM is funded by a Tier 2 Canada Research Chair in Mathematical Modeling and Program Science (CRC File Number 950-232643). The funders had no role in the study design, data collection and analysis, decision to publish, or preparation of the manuscript.

**Competing interests:** The authors have declared that no competing interests exist.

and disruptions in HIV service delivery, all of which contributed to delay in diagnosis and compromised treatment adherence.

## Background

Tuberculosis (TB) infects 1.2 billion people worldwide each year and caused an estimated 1.25 million deaths in 2023 [1,2]. Sierra Leone is one of the 30 high TB burden countries, which together account for 87% of the global TB burden. The TB incidence rate in Sierra Leone in 2023 was 273 per 100,000 population [2].

With tremendous progress in case notification by the country's national TB program in the past ten years, the number of confirmed TB cases increased from 13,195 in 2010–17,865 in 2019 [1]. In recent years, these figures have even improved further. Of the 24,000 TB cases expected in 2024, 22,381 (93%) were notified to the National TB Programme. Several challenges in TB care affect sustainability of these gains. The need for prolonged TB treatment increases the risk of treatment interruptions, leading to missed cases and the development of resistance to the most effective TB drugs, such as rifampicin and isoniazid [3,4].

In 2020, an estimated 101,000 cases of drug-resistant TB were reported in Africa [1]. In Sierra Leone, the prevalence of drug-resistant TB was estimated to be 4.3 per 100,000 population in 2023, including 2.8% of newly diagnosed cases and 21% of retreated patients [2]. This huge burden of drug-resistant TB poses additional challenges to TB service delivery in the country.

The early stages of the COVID-19 pandemic portray its adverse impact on adherence to drug-resistant TB treatment, which was exacerbated by movement restrictions that disrupted health systems and implementation of TB services [5,6]. Subsequent waves of the pandemic potentially increase TB transmission, delay detection, thereby exacerbating challenges and leading to poorer treatment outcomes and increased drug resistance [7–9].

To date, no studies have evaluated the impact of COVID-19 on treatment outcomes for drug-resistant TB in Sierra Leone. Global evidence is limited to only 6 articles on PubMed using the search term 'impact of COVID-19 AND treatment outcomes of drug resistance TB' on October 29, 2024, and none reported data specifically on drug resistant TB treatment outcomes [10–15]. Understanding the impact of COVID-19 on drug-resistant TB treatment outcomes has important policy implications for safeguarding essential health services in future public health emergencies.

This study aims to document the impact of COVID-19 on drug-resistant TB services in Sierra Leone by describing and evaluating the demographics, clinical characteristics, and outcomes of newly registered drug-resistant TB cases before, during, and after COVID-19.

## Methods

### Ethics statement

Ethical approval was obtained from the Sierra Leone Ethics and Scientific Review Committee with approval number 017/05/2023. Routine data was abstracted under a

waiver of informed consent from the ethics committee and approval from the health facility management. The authors did not have access to information that could identify individual participants during or after data collection.

**Study design.** We conducted a retrospective cohort study to examine newly registered drug-resistant TB cases between January 1, 2017, and December 31, 2022. Individual patient data were extracted from the National drug-resistant TB database.

**Study setting.** We extracted patient-level data of adult (aged 18 years or older) with drug-resistant TB attending the National TB Referral Hospital (Lakka Government Hospital), a 150-bed hospital in western Sierra Leone. As the country's primary center for the intensive phase treatment of drug-resistant TB, it plays a central role in managing complex TB cases. The hospital regularly admits patients with drug-sensitive TB and drug-resistant TB for intensive phase treatment as recommended by the national and World Health Organization (WHO) treatment guidelines. After discharge, the patients receive treatment during the continuation phase as outpatient and are closely followed up with medication refills, laboratory tests, and clinical evaluations until they are cured. The hospital has six doctors, 90 nurses, 19 laboratory scientists/technicians, and four radiographers.

In October 2020, Sierra Leone started using a standardized short multi-drug resistant-TB regimen to replace the long conventional regimens. In 2022, bedaquiline, pretomanid, linezolid and moxifloxacin (BPaL-M) and bedaquiline, pretomanid, linezolid (BPaL) (if fluoroquinolone-resistant) were adopted to treat multi-drug-resistant TB/rifampicin resistant-TB for all cases aged 14 years or older in the other treatment centre, but were rolled out to Lakka Government Hospital in 2023. [16].

Patient records from this hospital are routinely submitted to the National Drug-Resistant TB Database.

**Study population.** Data on drug-resistant TB/rifampicin-resistant (RR)-TB patients were obtained from the National drug-resistant TB Database. A total of 739 cases were registered, but we excluded 38 patients still receiving treatment and patients with missing variables, leaving a total of 701 cases for outcome analysis.

**Cohort classification and outcome analysis.** To assess the impact of the COVID-19 pandemic on treatment outcomes, we grouped patients into three cohorts according to their treatment initiation dates: the pre-COVID-19 cohort started treatment between January 2017 and February 2020. The COVID-19 cohort includes patients initiated on treatment between March 2020 and February 2022. The post-COVID-19 cohort includes patients who started treatment after the COVID-19 emergency phase, between March and December 2022. Treatment outcomes—including cured, loss to follow-up, and mortality—were compared across these cohorts to identify trends and disruptions associated with the pandemic.

**Data sources and variables.** Deidentified patient-level data of Lakka Government Hospital were extracted from the National Drug-resistant TB Database on September 24, 2023 in accordance with national data protection guidelines and ethical standards for secondary data analysis. We collected data on age, gender, body mass index (BMI), HIV status, TB treatment status (new or retreatment), treatment regimen (short vs. long) and treatment outcomes {successful (cured, treatment completed) and unsuccessful (death, loss to follow-up, treatment failure and not evaluated)} [17]. TB treatment outcomes were defined in accordance with the WHO reporting framework for patients treated for rifampicin resistant -TB, multi-drug-resistant TB and extensive drug-resistant TB using second-line TB medicines (Table 1).

**Analysis.** Data was analysed using STATA. Descriptive analysis was used to summarise the data on the demographic features, clinical characteristics and the treatment outcomes of drug-resistant- TB cases using proportion. We used logistic regression to examine the association between time-period and treatment outcomes (successful vs. not successful in the pre, during and post-COVID era), after adjusting for potential confounders (age, gender, nutritional status, HIV status and treatment regimens). We reported crude odds ratios (OR) and adjusted odds ratios (aOR) accordingly. A p-value of <0.05 was considered statistically significant at 95% confidence level. Model fit was assessed using the Hosmer-Lemshow test.

**Model diagnostics and missing data.** We employed standard diagnostic tools to assess the robustness of our logistic regression model. We did not encounter multicollinearity issues, as all Variance Inflation Factors were less than

**Table 1. Outcomes for Rifampicin resistant-TB/Multi-drug resistant-TB patients treated using second-line treatment.**

| Outcome | Definition |
|---|---|
| Cured | Treatment completed as recommended by the national policy without evidence of failure AND three or more consecutive cultures taken at least 30 days apart are negative after the intensive phase. |
| Treatment complete | Treatment completed as recommended by the national policy without evidence of failure BUT no record that three or more consecutive cultures taken at least 30 days apart are negative after the intensive phase. |
| Treatment failed | Treatment terminated or need for permanent regimen change of at least two anti-TB drugs because of:<br>• lack of conversion by the end of the intensive phase or<br>• bacteriological reversion in the continuation phase after conversion to negative or<br>• evidence of additional acquired resistance to fluoroquinolones or second-line injectable drugs or<br>• adverse drug reactions |
| Died | A patient who dies for any reason during the course of treatment |
| Lost to follow up | A patient whose treatment was interrupted for two consecutive months or more |
| Not evaluated | A patient for whom no treatment outcome is assigned. This includes cases "transferred out" to another treatment unit and whose treatment outcome is unknown |
| Treatment success | The sum of cured and treatment completed |
| Unsuccessful treatment outcome | The sum of death, failure, loss to follow-up, and not evaluated |

2.0. The final model demonstrated a good fit, with a Hosmer-Lemeshow p-value of 0.62, and acceptable discrimination, as indicated by a C-statistic of 0.72.

Of the 739 registered cases, 38 (5.1%) were excluded due to ongoing treatment or missing outcome data. The Little's Missing Completely at Random (MCAR) test indicated that missingness was random ((p = 0.12). Sensitivity analyses using best- and worst-case scenarios confirmed that these exclusions did not affect the study's main findings.

To account for potential confounding and temporal trends, we made additional adjustments for regimen type and calendar year, considering the major policy shift in October 2020 that introduced standardized short-course multi-drug-resistant TB regimens during the COVID-19 pandemic. We also tested for effect modification by including an interaction term between treatment period and regimen type.

## Results

### Demographic and clinical profile of registered patients with drug-resistant TB

Of the 701 drug-resistant TB patients, 383 (54.6%) were registered in the pre-COVID-19 period, 228 (32.5%) during COVID-19, and 92 (12.8%) in the post-COVID-19 period. Nearly three-fourths (284, 73.2%) of patients were aged 15–44 years. The patients' mean ages were 10.4 years (SD = 3.3) for those between 0 and 14 years old, 29.9 years (SD = 7.3) for those between 15 and 44 years old, and 52.3 years (SD = 7.3) for those 45 years and older.

Males comprised a larger proportion (523, 68.5%) of patients with drug-resistant TB in all periods. However, the proportion of women with drug resistant TB increased from 25.8% in the pre-COVID-19 period to 33.6% during COVID-19 and 31.5% in the post-COVID-19 period. The re-treatment TB cases reduced from 359 (92.5%) in the pre-COVID-19 period to 80 (30.9%) in the COVID-19 era. On the other hand, new treatment cases increased from 29 (7.5%) to 159 (61.4%) during the COVID-19 period (Table 2).

**Table 2. Demographic and clinical profile of registered patients with drug-resistant TB.**

| Demographics/ Clinicals | Sub-groups | Study periods | | | | | |
|---|---|---|---|---|---|---|---|
| | | Pre-COVID-19 N = 383 | | COVID-19 N = 228 | | Post-COVID-19 N = 90 | |
| | | No. | % | No. | % | No. | % |
| Age groups | 0 – 14 Years | 8 | 2.1 | 2 | 0.8 | 2 | 2.2 |
| | 15 – 44 Years | 284 | 73.2 | 204 | 78.8 | 65 | 70.7 |
| | ≥ 45 Years | 76 | 24.7 | 53 | 20.5 | 25 | 27.2 |
| Gender | Male | 288 | 74.2 | 172 | 66.4 | 63 | 68.5 |
| | Female | 100 | 25.8 | 87 | 33.6 | 29 | 31.5 |
| Nutrition status | Malnourished | 255 | 65.7 | 166 | 64.1 | 65 | 70.7 |
| | Not malnourished | 133 | 34.3 | 93 | 35.9 | 27 | 29.3 |
| HIV status | Positive | 68 | 17.5 | 69 | 26.6 | 20 | 21.7 |
| | Negative | 320 | 82.5 | 190 | 73.4 | 72 | 78.3 |
| Regimen | Short | 364 | 93.8 | 109 | 42.1 | 30 | 32.6 |
| | Longer | 24 | 6.2 | 150 | 57.9 | 62 | 67.4 |
| Drug-resistant TB type* | New | 29 | 7.5 | 159 | 61.4 | N/A | N/A |
| | Retreatment | 359 | 92.5 | 80 | 30.9 | N/A | N/A |
| | Missing | 00 | 0.0 | 20 | 7.7 | 92 | 100.0 |

Pre-COVID: January 1, 2017, to December 2022; during COVID-19 emergency: March 2020 to February 2020 post-COVID-19 emergency: March to December 2022. *N/A (not available): there was no information recorded on the type of TB among patients recorded during the post-COVID-19 period; Column percentages, Abbreviations: HIV- Human Immunodeficiency Virus; DR (Drug Resistant- Tuberculosis);; MDR-Multi-drug resistant.

Malnutrition among drug resistant TB patients increased from 65.7% in the pre-COVID-19 period to 70.7% in the COVID-19 era. HIV positivity among drug-resistant TB cases increased from 17.5% in the pre-COVID-19 period to 26.6% during COVID-19 and 21.7% in the COVID-19 period (Table 2).

### Treatment outcomes of patients with drug-resistant TB

Treatment outcomes differed across the study periods. The proportion of patients with drug-resistant TB who completed treatment decreased from 74.7% in the pre-COVID-19 period to 63.3% during COVID-19 and 68.5% post-COVID-19. The number of deaths among patients with drug-resistant TB was higher during the COVID-19 period (22.8%) and post-COVID period (20.0%) than in the pre-COVID period (16.7%). Similarly, more patients with drug-resistant TB were lost follow up in the post-COVID-19 period (5.6%) and COVID-19 period (4.4%) than in the pre-COVID-19 period (4.2%). There were more patients with successful treatment outcomes before the COVID-19 pandemic (74.7%) than during COVID-19 (63.3%) and after COVID-19 period (70.0%) (Table 3).

### Factors associated with successful drug-resistance TB treatment outcomes

Using logistics regression analysis, we investigated the association between the COVID-19 pandemic and TB treatment outcomes. In the univariable analysis, we found no significant association between the COVID-19 period and TB treatment outcomes, using the pre-COVID period as reference ($P = 0.822$; $OR$=0.300; $95\%$ $CI$: 0.567- 1.191).After adjusting for potential confounders (age, gender, nutritional status, HIV status, and treatment regimens), there was no significant association between the study period and treatment outcomes [$aOR$: COVID-19 = 1.150($95\%$ $CI$: 0.723 – 1.829); post-COVID-19 = 1.202($95\%$ CI: 0.657– 2.199)]. Patients who were malnourished had higher odds of successful treatment outcomes (aOR: 1.482, $95\%$ CI: 1.007 –2.183), whereas those on short regimen had lower odds of successful treatment

**Table 3. Clinical characteristics and the treatment outcomes of newly registered drug-resistant TB cases pre-COVID-19, COVID-19 and post-COVID-19.**

| Treatment outcomes | Sub-groups | Study periods | | | | | |
|---|---|---|---|---|---|---|---|
| | | Pre-Covid N=383 | | Covid N=228 | | Post-Covid N=90 | |
| | | No. | % | No. | % | No. | % |
| All treatment outcomes | Complete | 290 | 75.7 | 164 | 71.9 | 63 | 70.0 |
| | Died | 64 | 16.7 | 52 | 22.8 | 18 | 20.0 |
| | Failed | 13 | 3.4 | 2 | 0.9 | 4 | 4.4 |
| | LTFU | 16 | 4.2 | 10 | 4.4 | 5 | 5.6 |
| Outcomes (Binary) | Successful | 290 | 75.7 | 164 | 71.9 | 63 | 70.0 |
| | Unsuccessful | 93 | 24.3 | 64 | 28.1 | 27 | 30.0 |

* Unsuccessful treatment outcome includes death, failure, and loss to follow-up (LTFU). It does not include "on treatment".

outcome (aOR: 0.51, 95% CI: 0.321–0.810) compared to the longer regimen. There was no significant association between HIV status and TB treatment outcomes in multivariable analysis (aOR: 1.341; 95% CL=0.874 - 2.056) (Table 4).

## Discussion

The study provides a comprehensive assessment of the impact of the COVID-19 pandemic on drug-resistant TB services in Sierra Leone, focusing on demographic and clinical characteristics and treatment outcomes of newly registered drug-resistant TB cases across the different pandemic periods. Utilizing programmatic data, the study reported a significant effect of COVID-19 on drug-resistant TB notifications, and patient profile. There were more cases of drug-resistant TB during the COVID-19 pandemic than before the pandemic. Second, there were gender and age disparities, with more cases of drug-resistant TB among young male population. Malnutrition and HIV positivity among drug resistant TB patients

**Table 4. Factors associated with drug-resistant TB treatment outcomes in Sierra Leone.**

| Variable | Sub-variables | OR | 95% CI | | P-value | aOR | 95% CI | | P-value |
|---|---|---|---|---|---|---|---|---|---|
| | | | Lower | Upper | | | Lower | Upper | |
| Study period | Pre-Covid | Ref | | | | Ref | | | |
| | Covid | 0.3 | 0.57 | 1.19 | 0.80 | 1.15 | 0.72 | 1.83 | 0.56 |
| | Post-Covid | 0.26 | 0.45 | 1.20 | 0.70 | 1.20 | 0.66 | 2.19 | 0.55 |
| Gender | Male | | | | | Ref | | | |
| | Female | | | | | 1.14 | 0.76 | 1.705 | 0.53 |
| Age groups | 0 – 14 Years | | | | | Ref | | | |
| | 15 – 44 Years | | | | | 2.10 | 0.63 | 7.04 | 0.23 |
| | ≥ 45 Years | | | | | 0.99 | 0.29 | 3.43 | 0.99 |
| Nutritional status | Malnourished | | | | | Ref | | | |
| | Not malnourished | | | | | 1.48 | 1.01 | 2.18 | **0.046** |
| Ent HIV status | Positive | | | | | Ref | | | |
| | Negative | | | | | 1.34 | 0.874 | 2.06 | 0.18 |
| Treatment regimen | Short | | | | | Ref | | | |
| | Longer | | | | | 0.51 | 0.32 | 0.81 | **0.004** |

Bold figures represent significant findings.

increased following the COVID-19 pandemic. Finally, there were more patients with successful treatment outcomes before the COVID-19 pandemic (74.7%) than during COVID-19 (63.3%); however, this difference was likely influenced by the simultaneous introduction of newer, more effective regimens and changes in the patient population, including nutritional profile and HIV infection.

The marked increase in the number of notified drug-resistant TB cases, particularly among people aged 15–44 years, may be explained by the living conditions of young people in poor, overcrowded housing facilities with limited access to healthcare and increase their risk of contracting TB [18]. Movement restrictions and lockdowns during the COVID-19 pandemic may have exacerbated overcrowding and increased the risk of TB infection. In Iraq, there was a significant increase in drug-resistant TB cases among young people aged 15–34 years, similar to our findings [19]. Young people engage in many activities, such as smoking, and have inherent immunopathology and hormonal changes that put them at risk for TB infection [19].

There has been a shift in the gender distribution trend, with an increase in the proportion of male drug-resistant TB cases during the pandemic period. These demographic shifts may underscore the pandemic's impact on disease surveillance and patient demographics [20] and call for mainstreaming of gender-related interventions in public health emergency preparedness and response in low-income countries.

COVID-19 disrupted health services delivery and supply chain of essential health commodities, leading to poor health outcomes [21]. This study highlighted adverse changes in clinical parameters during the pandemic. The recovery phase of the COVID-19 pandemic saw a notable rise in malnutrition among drug-resistant TB patients compared to pre-pandemic levels. Malnutrition is a public health problem in Sierra Leone [22]. The interaction of social factors such as poverty with COVID-19 and TB may explain the worsening malnutrition among patients with drug-resistant TB in this study. Poverty worsens during the COVID-19 and post-COVID-19 period [23]. Furthermore, restrictions during the pandemic reduced the quality and productivity of plants, meat and eggs, which worsens malnutrition [24].

There was an increased proportion of HIV-positive cases with drug-resistant TB in the post-pandemic period compared to the pre-pandemic period. This finding reflects the higher likelihood of a positive HIV test during the COVID-19 pandemic observed in a previous study in Sierra Leone [25]. The reason for the increased burden of HIV during and after the pandemic could be due to disruptions in HIV service delivery resulting in late presentation for care during and after the pandemic [26].

The COVID-19 period also witnessed an uptick in new TB cases and fluctuations in treatment regimens, reflecting the healthcare system's adaptive responses and challenges during crisis periods [1,2].

Other consequences of the COVID-19 restriction measures are the adverse effects on TB treatment outcomes. Treatment outcomes exhibited variability across the study periods, with a decline in treatment completion rates, increased treatment failures, and a high proportion of deaths observed during the phase of COVID-19. A similar study done on treatment outcomes in the country indicated that the lockdown imposed by the authorities on mobility and strikes by healthcare workers contributed to missed drug doses, leading to treatment failures and possible deaths [9].

There were more patients with successful treatment outcomes before the COVID- than during the COVID-19, but this apparent negative impact of the pandemic is attenuated on multivariable analysis, suggesting that the observed decline in the raw treatment success rate was due to pre-existing trends and health system adaptation coinciding with the pandemic. Nonetheless, the pandemic's indirect consequences, such as worsening malnutrition and increased rate of HIV co-infection point to deeper systemic vulnerabilities, including weak social protection mechanisms, increased food insecurity, and disruptions in HIV service delivery, all of which contributed to delay in diagnosis and compromised treatment adherence.

Our study has limitations. The study's analysis of retreatment cases in 2022 was limited by missing clinical variables, potentially influencing the findings. This could compromise the interpretation of the results, necessitating future research to ensure the completeness and accuracy of clinical data. Future research should prioritize data integrity for a better understanding of pandemic dynamics and optimizing outcomes for drug-resistant TB patients in low-income countries.

## Conclusion

We observed a decline in the drug-resistant TB treatment success outcomes during the COVID-19 period, which was primarily influenced by concurrent shifts in treatment protocols and underlying secular trends. The pandemic itself did not emerge as an independent determinant of poor treatment outcomes, highlighting the resilience of the TB care system.

Nonetheless, the pandemic's indirect consequences, such as worsening malnutrition and increased rate of HIV co-infection point to deeper systemic vulnerabilities, including weak social protection mechanisms, increased food insecurity, and disruptions in HIV service delivery, all of which contributed to delay in diagnosis and compromised treatment adherence.

## Acknowledgments

This research was conducted through a partnership between Sustainable Health Systems Sierra Leone, and the Li Ka Shing Research Institute, Unity Health, University of Toronto, Canada. The training model used that resulted in this publication was adapted from the Structured Operational Research and Training Initiative (SORT IT), a global partnership led by the Special Programme for Research and Training in Tropical Diseases at the World Health Organization (WHO/TDR, Geneva, Switzerland), for which SL, AKC and SM are SORT-IT Mentors. Mentorship was provided by Sustainable Health Systems, (Freetown, Sierra Leone); Ministry of Health, Government of Sierra Leone; University of Toronto (Toronto, Canada); Partners in Health Sierra Leone (Koidu, Sierra Leone). The authors would like to acknowledge Kristy Cheuk Yin Yiu and Bailah Molleh for Project Management and Coordination.

## Author contributions

**Conceptualization:** Josephine Amie Koroma, Mariama Mahmoud, Stephen Sevalie, Adrienne K. Chan, Sharmistha Mishra, Sulaiman Lakoh, Joseph Sam Kanu.

**Data curation:** Josephine Amie Koroma, Bailah Molleh, Joseph Sam Kanu.

**Formal analysis:** Josephine Amie Koroma, Adrienne K. Chan, Sharmistha Mishra, Joseph Sam Kanu.

**Funding acquisition:** Stephen Sevalie, Adrienne K. Chan, Sharmistha Mishra, Sulaiman Lakoh.

**Methodology:** Josephine Amie Koroma, Adrienne K. Chan, Sharmistha Mishra, Joseph Sam Kanu.

**Resources:** Stephen Sevalie, Adrienne K. Chan, Sharmistha Mishra, Sulaiman Lakoh.

**Supervision:** Stephen Sevalie, Adrienne K. Chan, Sharmistha Mishra, Sulaiman Lakoh.

**Validation:** Bailah Molleh, Joseph Sam Kanu.

**Writing – original draft:** Josephine Amie Koroma, Adrienne K. Chan, Sharmistha Mishra, Sulaiman Lakoh, Joseph Sam Kanu.

**Writing – review & editing:** Adrienne K. Chan, Sharmistha Mishra, Sulaiman Lakoh.

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
