## [Decision Letter · Decision Letter 0]

12 May 2025

PGPH-D-25-00640

Effect of the COVID-19 pandemic on drug-resistant tuberculosis treatment outcomes at a national referral hospital in Sierra Leone, 2017 to 2022: a retrospective study

Dear Dr. Lakoh,

Thank you for submitting your manuscript to PLOS Global Public Health. After careful consideration, we feel that it has merit but does not fully meet PLOS Global Public Health’s publication criteria as it currently stands. Therefore, we invite you to submit a revised version of the manuscript that addresses the points raised during the review process.

EDITOR:

This paper highlights the detrimental effects of the COVID-19 pandemic on the success of the tuberculosis program. After review by the reviewers and editor, the manuscript is deemed to have room for improvement. See the reviewers' recommendations below, as well as the comments in the attached file.

We look forward to receiving your revised manuscript.

Kind regards,

Dione Benjumea-Bedoya, Ph.D

Guest Editor

Journal Requirements:

Additional Editor Comments (if provided):

This paper highlights the detrimental effects of the COVID-19 pandemic on the success of the tuberculosis program. After review by the reviewers and editor, the manuscript is deemed to have room for improvement. See the reviewers' recommendations below, as well as the comments in the attached file.

Reviewers' comments:

Reviewer's Responses to Questions

**Comments to the Author**

1. Does this manuscript meet PLOS Global Public Health’s publication criteria?

Reviewer #1: Yes

Reviewer #2: Partly

2. Has the statistical analysis been performed appropriately and rigorously?

Reviewer #1: Yes

Reviewer #2: No

3. Have the authors made all data underlying the findings in their manuscript fully available (please refer to the Data Availability Statement at the start of the manuscript PDF file)?

Reviewer #1: Yes

Reviewer #2: No

4. Is the manuscript presented in an intelligible fashion and written in standard English?

Reviewer #1: Yes

Reviewer #2: Yes

Reviewer #1: I think it is a good article. One of the main comments is the elaboration of a table explaining the logistic regression variables with their respective adjusted ORs and confidence intervals. Plese review

Reviewer #2: The manuscript addresses an important public-health question—the impact of the COVID-19 pandemic on treatment outcomes in drug-resistant tuberculosis (DR-TB) in a high-burden country. Using the national DR-TB database (701 patients, 2017-2022), the authors compare programme success before, during and after the first COVID-19 wave. The work is potentially publishable in PLOS Global Public Health, but several methodological and reporting issues must be resolved before it can meet the journal’s publication criteria in full.

Major comments (required)

1. Study design is mis-classified. The manuscript calls itself a “retrospective cross-sectional study,” yet exposure (treatment period) clearly precedes outcome (end-of-treatment status). Re-define the work as a retrospective cohort (or time-series) study and update the Abstract, Methods, and STROBE checklist accordingly.

2. Confounding by treatment regimen and calendar time. Short, all-oral regimens (BPaL-M) were introduced mid-study. Because these regimens are strongly associated with outcome and correlate with the pandemic period, they can distort the effect estimate. Include regimen type as a potential effect modifier (interaction period x regimen) or stratify analyses. Add calendar year (or quarter) to control for secular trends.

3. Model diagnostics and robustness. The logistic model’s assumptions, goodness-of-fit and collinearity checks are not reported; 38 patients were excluded for missing data without sensitivity analyses. Report VIFs/condition indices, Hosmer–Lemeshow or equivalent, C-statistic, and residual plots. Describe the pattern of missingness and consider multiple imputation or best-/worst-case analyses.

4. Interpretation overstates the evidence. The crude OR is significant, but the adjusted OR (0.87; 95 % CI 0.57–1.33) is not. The Discussion still concludes that COVID-19 “significantly reduced” treatment success. Base all inferences on the adjusted model; acknowledge imprecision and possible residual confounding.

5. Data availability non-compliant. The Figshare link provided is private. Deposit a fully anonymised dataset and codebook under a public DOI. Ensure that every variable used in the analysis is included.

Minor comments (suggested)

1. Title vs. scope. The title mentions a “national referral hospital,” yet data derive from the National DR-TB Database covering multiple facilities. Please align title, setting and Methods.

2. Typographical errors. Correct “resitant,” “peroid,” and ensure “post-COVID-19” is used consistently.

3. Abbreviations. Define DR-TB, BPaL-M, aOR at first appearance; list all abbreviations in a separate section as per journal style.

Conclusion

The study addresses a timely question and uses a valuable national dataset, but the statistical approach, data-sharing compliance and interpretation require substantial revision. I would be pleased to review a thoroughly revised version.

**Do you want your identity to be public for this peer review?** For information about this choice, including consent withdrawal, please see our Privacy Policy

Reviewer #1: No

Reviewer #2: No

---

## [Decision Letter · Decision Letter 1]

23 Nov 2025

PGPH-D-25-00640R1

Effect of the COVID-19 pandemic on drug-resistant tuberculosis treatment outcomes at a national referral hospital in Sierra Leone, 2017 to 2022: a retrospective cohort study

Dear Dr. Lakoh,

Thank you for submitting your manuscript to PLOS Global Public Health. After careful consideration, we feel that it has merit but does not fully meet PLOS Global Public Health’s publication criteria as it currently stands. Therefore, we invite you to submit a revised version of the manuscript that addresses the points raised during the review process.

EDITOR:

It is important to highlight the effect of the COVID-19 pandemic on the treatment outcomes of drug-resistant tuberculosis, and the authors have addressed most of the reviewers' comments. However, some aspects remain to be addressed that could improve the work. Authors are encouraged to follow the reviewers' comments and submit a revision to continue the peer review process.

We look forward to receiving your revised manuscript.

Kind regards,

Dione Benjumea-Bedoya, Ph.D

Guest Editor

Journal Requirements:

Additional Editor Comments (if provided):

It is important to highlight the effect of the COVID-19 pandemic on the treatment outcomes of drug-resistant tuberculosis, and the authors have addressed most of the reviewers' comments. However, some aspects remain to be addressed that could improve the work. Authors are encouraged to follow the reviewers' comments and submit a revision to continue the peer review process.

Reviewers' comments:

Reviewer's Responses to Questions

**Comments to the Author**

Reviewer #1: All comments have been addressed

Reviewer #2: (No Response)

publication criteria?

Reviewer #1: Yes

Reviewer #2: Yes

3. Has the statistical analysis been performed appropriately and rigorously?

Reviewer #1: Yes

Reviewer #2: Yes

4. Have the authors made all data underlying the findings in their manuscript fully available (please refer to the Data Availability Statement at the start of the manuscript PDF file)?

Reviewer #1: Yes

Reviewer #2: No

5. Is the manuscript presented in an intelligible fashion and written in standard English?

Reviewer #1: Yes

Reviewer #2: Yes

Reviewer #1: I think it's a good article with the statistical rigor necessary for understand this important topic

Reviewer #2: Thank you for the opportunity to review the revised version of PGPH-D-25-00640_R1. The manuscript has improved substantially: the study is now correctly framed as a retrospective cohort, the statistical methods have been strengthened with additional adjustment for calendar year and regimen type and with appropriate model diagnostics and sensitivity analyses, and the interpretation of the association between the COVID-19 period and treatment outcomes has been suitably tempered. Overall, the work is methodologically sound, clearly reported and of clear relevance for global TB control. I have only minor remaining concerns, mainly relating to data sharing and small editorial details: the Figshare link provided in the Data Availability Statement still points to a private “account/edit” URL and therefore does not comply with PLOS’s open-data policy; I encourage the authors to deposit a fully anonymised dataset with an accessible public DOI (including a data dictionary and, if possible, analysis code) and to cite that DOI in the manuscript. In addition, I suggest a final consistency check to ensure that the design is uniformly described as a retrospective cohort throughout, that all abbreviations (e.g. DR-TB, BPaL-M, aOR) are defined at first use, and that minor typographical issues and formatting of percentages and confidence intervals are corrected. Once these small points—particularly the public data link—are addressed, I consider the manuscript suitable for publication in PLOS Global Public Health and recommend Minor Revision.

**Do you want your identity to be public for this peer review?** For information about this choice, including consent withdrawal, please see our Privacy Policy

Reviewer #1: No

Reviewer #2: No

---

## [Decision Letter · Decision Letter 2]

4 Feb 2026

Effect of the COVID-19 pandemic on drug-resistant tuberculosis treatment outcomes at a national referral hospital in Sierra Leone, 2017 to 2022: a retrospective cohort study

PGPH-D-25-00640R2

Dear Dr. Lakoh,

We are pleased to inform you that your manuscript 'Effect of the COVID-19 pandemic on drug-resistant tuberculosis treatment outcomes at a national referral hospital in Sierra Leone, 2017 to 2022: a retrospective cohort study' has been provisionally accepted for publication in PLOS Global Public Health.

Best regards,

Dione Benjumea-Bedoya, Ph.D

Guest Editor

The effect of the COVID-19 pandemic on drug-resistant tuberculosis treatment outcomes in an important topic, and the authors addressed every comment from reviewers.

Reviewer Comments (if any, and for reference):

Reviewer's Responses to Questions

**Comments to the Author**

Reviewer #2: All comments have been addressed

publication criteria?

Reviewer #2: Yes

3. Has the statistical analysis been performed appropriately and rigorously?

Reviewer #2: Yes

4. Have the authors made all data underlying the findings in their manuscript fully available (please refer to the Data Availability Statement at the start of the manuscript PDF file)?

Reviewer #2: Yes

5. Is the manuscript presented in an intelligible fashion and written in standard English?

Reviewer #2: Yes

Reviewer #2: Thank you for the careful and thorough revision of your manuscript. In this R2 version you have fully addressed the previous major and minor comments: the study is consistently framed as a retrospective cohort, the statistical methods and diagnostics are clearly presented, the interpretation of the association between COVID-19 periods and treatment outcomes is appropriately cautious, and the data are now shared through an open and citable Figshare DOI. Overall, the manuscript is methodologically sound, clearly written and highly relevant for TB control in high-burden settings. I have no further substantive comments and support the manuscript for publication in PLOS Global Public Health.

**Do you want your identity to be public for this peer review?** For information about this choice, including consent withdrawal, please see our Privacy Policy

Reviewer #2: No
